# A Wideband and Low-Power Distributed Cascode Mixer Using Inductive Feedback

**DOI:** 10.3390/s22229022

**Published:** 2022-11-21

**Authors:** Jihoon Kim

**Affiliations:** Department of Electronics Engineering, Pai Chai University, Daejeon 35345, Republic of Korea; j7h7.kim@pcu.ac.kr; Tel.: +82-42-520-5595

**Keywords:** wideband, low power, distributed, cascode mixer, gate positive feedback, CMOS

## Abstract

A wideband and low-power distributed cascode mixer is implemented for future mobile communications. The distributed design inspired by the distributed amplifier (DA) enables a mixer to operate in a wide band. In addition, the cascode structure and inductive positive feedback design allow high conversion gain with low-power consumption. The proposed mixer is fabricated using a 130 nm commercial complementary metal-oxide-semiconductor (CMOS) process. It consists of three cascode gain cells and operates with a drain voltage of 1.5 V and a gate voltage of 0.5 to 0.7 V. The fabricated mixer exhibits conversion gain of −2.9 to 3.1 dB at the radio frequencies (RFs) of 4 to 30 GHz and −1.9 to 0.4 dB at RFs of 54 to 66 GHz under the conditions of 8 to 10 dBm of local oscillator (LO) power and 650 MHz of intermediate frequency (IF). The LO-RF isolation is more than 15 dB over the entire measurement band (0.2 to 67 GHz) as the RF and LO signals are applied to different transistors owing to the cascode structure. The total power consumption is only within 12 mW, and the chip size is 0.056 mm^2^, making it possible to implement a compact mixer. The proposed mixer shows broadband characteristics covering from ultra-wideband (UWB) and the 28 GHz fifth-generation (5G) communication band to the 60 GHz wireless gigabit alliance (WiGig) band.

## 1. Introduction

A mixer is a key component that converts the radio frequency (RF) at the frontend of a communication system. At the transmitter, the mixer up-converts the frequency of a signal from an intermediate frequency (IF) to an RF, and at the receiver, it down-converts the frequency in reverse. Recently, with the commercialization of fifth-generation (5G) communication, many RF chains have been integrated into an antenna module as array antennas have been adopted for beamforming purposes [1,2]. Figure 1 shows a block diagram of a 5G antenna module commonly used in a mobile handset or base station. As shown in Figure 1, it is important to reduce power consumption as there are many chains due to the array type. This requirement applies to not only power amplifiers, which traditionally account for a significant portion of power consumption, but also receivers that must always be waiting to receive data. In addition, as various frequencies are used for many communication services, many transmitter/receiver modules operating at each frequency are required. Figure 2 shows the frequency spectrum distribution in commercial use today [3]. As the demand for mobile handsets increases, it has come to a situation in which a single user equipment has to support various communication services. Therefore, the demand for a wideband transceiver capable of covering multiple frequency bands with a single module or integrated circuit is increasing. For example, almost all Wi-Fi devices used in homes or public places now have a built-in chip capable of 2.4 GHz/5 GHz dual-band communication. 

A typical type of broadband amplifier is a distributed amplifier. Usually, the bandwidth of an RF circuit is governed by a transistor, and the bandwidth of a transistor is limited by the resistances, capacitances, and inductances seen in the equivalent circuit of the transistor [4]. As the frequency increases, the gain of the transistor decreases rapidly due to their influence. Distributed amplifiers are contrived by taking the idea from the structure of a transmission line having a wide bandwidth [5]. They are designed to absorb the gate/drain resistances and capacitances inside the transistors by arranging transistors in a row. Therefore, although the gain is relatively small compared with the number of transistors used due to inefficient transistor arrangement, it is possible to obtain a bandwidth of more than an octave band [6,7,8,9,10,11].

As the demand for wideband transceivers increases, research studies to obtain wideband characteristics by adopting a distributed amplifier design to not only amplifiers such as power amplifiers (PAs) and low noise amplifiers (LNAs), but also other RF components such as mixers, doublers, switches, and true-time delay circuits are being actively conducted [12,13,14,15,16,17,18]. 

In line with this trend, a wideband mixer operating from several gigahertz to the V-band by applying the distributed design method is implemented. Field effect transistor (FET) mixers widely used today are resistive mixers, Gilbert cell mixers, and drain mixers [19,20,21]. The resistive mixer uses a linear region near a zero drain bias of a FET, and it consumes no dc power and has excellent linearity [19]. However, there is a disadvantage in that the conversion loss is large. On the other hand, the Gilbert cell mixer is a circuit that implements a current multiplier and can obtain a high conversion gain and has excellent isolation, but consumes a lot of dc power [20]. The drain mixer is an intermediate feature between the resistive mixer and the Gilbert cell mixer, and uses the nonlinearity of the drain current near the knee voltage [21]. A positive conversion gain (CG) can be obtained by supplying an RF signal to the gate, and dc power consumption is not large compared to Gilbert cell mixers. This study applies a distributed structure and a cascode structure to this drain mixer. 

A number of distributed mixers have been reported so far, but most of them are based on Ⅲ−Ⅳ semiconductors, and most of the CMOS mixers that can be used in silicon-based system-on-chip (SoC) today have a high DC power consumption or a low CG [12,13,22,23,24,25,26]. In this study, by designing a unit mixer cell as a cascode type with inductive feedback, a high CG is obtained with a low-power consumption.

## 2. Circuit Design

### 2.1. Bias Selection of Cascode Drain Mixer

The cascode structure is composed of a combination of a common source transistor and a common gate transistor. The common source transistor functions as a mixer, and the common gate transistor amplifies an IF signal generated as a result of mixing [27,28]. There are many ways to operate a transistor as a mixer, but a drain mixer that supplies an RF signal to the gate port and an LO signal to the drain port to extract an IF signal is widely used because it is convenient to implement in a wideband distributed design [12,29]. Moreover, since a large LO power can be used as a drain bias, it has the advantage of low-power consumption. Therefore, even in the cascode structure, the common source transistor is implemented as a drain mixer. In this case, finding a suitable gate-to-source voltage (V_gs_) is critical to the performance of the mixer. As mentioned in [12], the IF current of a typical CMOS drain mixer is generally proportional to the trans-conductance (g_m_). More precisely, it is proportional to the derivative of g_m_ with respect to the drain-to-source voltage (V_ds_). Figure 3 shows the simulated derivative of g_m_ with respect to V_ds_ according to V_gs_ and V_ds_. As shown in Figure 3, ∂g_m_/∂V_ds_ is maximum when V_gs_ is 0.7 V near V_ds_ = 0.15 V. 

Figure 4a,b show the circuit schematic of the cascode mixer and DC-IV of each transistor constituting the cascode structure, respectively. As shown in Figure 4a, each transistor constituting the cascode cell should share the same current under a fixed drain voltage (V_DD_). In Figure 3, the bias of M1 favorable to designing the drain mixer was V_ds_ = 0.15 V and V_gs_ = 0.7 V. Therefore, even if the total V_DD_ is 1.5 V, it is sufficient for M2 to operate as an IF amplifier.

When V_DD_ is set to 1.5 V, the DC-IV of M1 and M2 is displayed together as shown in Figure 4b. If we mark the point where the two DC-IV curves meet so that the bias of M1 becomes V_ds_ = 0.15 V and V_gs_ = 0.7 V, we can see that it corresponds to the circle indicated by the black dotted line in Figure 4b. As a result, through the above analysis, the bias of the cascode mixer can be set as V_DD_ = 1.5 V, V_GG2_ = 0.7 V, and V_GG1_ = 0.7 V.

### 2.2. Selection of Transistor Size and Stage Number 

To determine the transistor size and stage number of a cascode mixer, the operating frequency range must be considered. The operating frequency range is mainly limited by the parasitic capacitances and resistances of the transistor seen at the gate line [5]. When the transistor size and stage number are increased, CG is usually improved as g_m_ is increased, but the operating frequency range is decreased because the parasitic capacitances inside the transistor are increased. Therefore, it is important to select the appropriate transistor size and stage number.

For the mixer simulation, the BSIM4 RF CMOS model provided by the foundry is basically used [30]. But when the S-parameters of the foundry model were compared to the measured data, it was still inaccurate at high frequencies. We composed a macro model by adding external parasitic parameters in the BSIM4 RF model. Figure 5 shows the macro model of RF CMOS used in this work. Using the macro model, S-parameters and a drain current at active bias are simulated for each transistor size. Figure 6a,b show the simulated S11 and S21 on the Smith chart. The smaller the size of the transistor, the smaller the power consumption, and the smaller the input parasitic capacitance, so that a high-frequency mixer operation is possible. However, since the g_m_ becomes small, it is disadvantageous to the conversion gain. Therefore, it is necessary to determine the size of the transistor by comprehensively considering the maximum operating frequency, conversion gain, and power consumption. The stage number can be analyzed using common-source gain cells similarly to the method proposed in [12]. Table 1 compares the simulated 3-dB cutoff frequency (F_cut-off_) according to the stage number. 

From the above results, the size of the mixer transistor and the stage number of the mixer are determined to obtain the bandwidth up to V-band. Thus, the size of the mixer transistor is set to 30 μm and the stage number of the mixer is set to 3.

### 2.3. Design of Gate Line Inductance and Inductive Positive Feedback Inductance

After determining the transistor size of the cascode cell and the number of stages of the distributed mixer, it is necessary to determine the line length to connect each mixer cell. In the distributed mixer design, the gate line length and drain line length must be determined, but the first priority is to determine the gate line length, which has more influence on the bandwidth. When the gate line length is determined, the drain line length is determined so that the phase of the signal passing through the gate line and that of the signal passing through the drain line are the same in order to cancel the reverse gain according to the distributed amplifier theory [4].

Figure 7a,b show the simulated magnitudes of (a) S21 and (b) S11 in the total gate lines of a distributed cascode mixer versus RF according to gate line inductance (L_g_), respectively. Because the gate/drain lines are usually implemented as transmission lines, they can be expressed as an R-L-G-C model [4]. Since a narrow and long transmission line is used as the gate line, the line inductance (L) becomes dominant among R-L-G-C parameters. Assuming there is no loss and the line capacitance (C) is absorbed by the input capacitance of the transistor, the gate line length can be modeled as a simple inductance.

Moreover, as determined in the previous section, the transistor size is fixed to 30 μm and the total number of stages is fixed to three. As shown in Figure 7a, as the line inductance increases, the loss increases at high frequencies, so the F_cut-off_ decreases from 96 GHz when L_g_ is 50 pH to 68 GHz when L_g_ is 150 pH. However, Figure 7b shows that as the line inductance increases, it is combined with the parasitic capacitances of the transistor, and the RF return loss improves further in the broadband. When L_g_ is 30 pH, it is about 7 dB at 92 GHz, but when L_g_ is 150 pH, it increases to about 18.4 dB at 92 GHz. Therefore, it is necessary to select an appropriate L_g_ value considering both F_cut-off_ and RF return loss. In this study, 120 pH is selected as the gate line inductance, and the gate line length and drain line length are designed accordingly. 

To further improve the CG at high frequencies, positive feedback using an inductor can be applied to the cascode structure [10,31]. In the cascode structure, the positive feedback can be divided into two types. One is adding an inductor to the gate of the upper transistor, and the other is adding an inductor to the source-drain connection between transistors. The inductors used here help improve the bandwidth by suppressing the decrease in g_m_ of the entire cascode cell at high frequencies. For each of the two cases, we will analyze them mathematically using the equivalent circuit of the cascode structure.

Figure 8a,b show the transistor equivalent circuits of the cascode mixer when the first positive feedback inductor (L_p1_) is added to the gate node of the upper transistor (M2). R_gs_ is the gate-source resistance, C_gs_ is the gate-source capacitance, C_gd_ is the gate-drain capacitance, C_ds_ is the drain-source capacitance. The C_gd_ value is generally small (less than 20 fF), and since the size of the transistors used in the actual analysis is small, the C_gd_ value is smaller. Therefore, it can be omitted as shown in Figure 8b. Since the RF signal is mixed with the LO signal at the output terminal of M2 and converted to the IF output, it is reasonable to analyze the effect of power gain on the RF signal. To know power gain of the RF signal, it is necessary to calculate S21, and by the Z-to-S conversion equations, S21 is
(1)$$S21=\frac{2\cdot Z_{21}\cdot Z_{0}}{\left[\left(Z_{11}+Z_{0}\right)\left(Z_{22}+Z_{0}\right)\right]}$$
where *Z*_0_ is the characteristic impedance [4]. 

In this study, since *Z*_11_ and *Z*_22_ are generally much larger than *Z*_0_ and do not change significantly according to the feedback inductance, only *Z*_21_ is calculated to simplify the formula and compared the cases with and without inductive feedback. The *Z*_21_ calculated through node analysis of the equivalent circuit in Figure 8b is
(2)$$Z_{21}=\frac{P\cdot g_{m}\cdot Z_{1}}{sC_{ds}\left[2\left(1-\frac{sC_{ds}}{T}\right)-\frac{1}{Z_{2}T}\right]}=\frac{P\cdot g_{m}\cdot Z_{1}}{2sC_{ds}-\frac{s^{2}C_{ds}^{2}}{T}-\frac{1}{Z_{2}T}}\;\approx \frac{P\cdot g_{m}\cdot Z_{1}}{2sC_{ds}-\frac{s^{2}C_{ds}^{2}}{T}-\frac{sC_{ds}}{T}}$$
(3)$$P=\frac{1}{1+sR_{gs}C_{gs}}$$
(4)$$Z_{1}=R_{gs}+\frac{1}{sC_{gs}}$$
(5)$$Z_{2}=\left(sL_{p1}+R_{gs}+\frac{1}{sC_{gs}}\right)//\frac{1}{sC_{ds}}$$
(6)$$T=\frac{g_{m}}{1+sR_{gs}C_{gs}+s^{2}L_{p1}C_{gs}}+sC_{ds}$$
where *s* = *j* × *ω* and *ω* = 2 × *π* × frequency. 

In Equation (2), expressions related to L_p1_ are included in the denominator term. Since C_ds_ is a very small value, 1/*Z*_2_ can be approximated with sC_ds_. As shown in Equation (6), L_p1_ is added to the denominator term of *T*, resulting in a negative real term and the magnitude of T increases. Accordingly, it can be seen that the denominator term of *Z*_21_ decreases the real part, so that the magnitude of the denominator becomes small, and the magnitude of *Z*_21_ increases, so that power gain is also improved. However, when L_p1_ becomes too large, the real part of the denominator of T increases again and the magnitude of *Z*_21_ decreases, so that power gain decreases. Therefore, it is important to choose an appropriate L_p1_ value between the two cases.

Figure 9a,b show the transistor equivalent circuits of the cascode mixer when the second positive feedback inductor (L_p2_) is added to the source-drain connection between M1 and M2. In a similar way as before, the *Z*_21_ calculated through node analysis of the equivalent circuit in Figure 9b is
(7)$$Z_{21}=\frac{\left(g_{m}P+sC_{ds}\right)P\cdot g_{m}\cdot Z_{1}}{sC_{ds}\left[\left(1+s^{2}L_{p2}C_{ds}\right)\left(2\cdot g_{m}P-\frac{1}{Z_{2}}\right)-sC_{ds}\right]}$$
(8)$$P=\frac{1}{1+sR_{gs}C_{gs}}$$
(9)$$Z_{1}=Z_{2}=R_{gs}+\frac{1}{sC_{gs}}$$

In Equation (7), expressions related to L_p2_ are included in the denominator term. When L_p2_ is added, the $1+s^{2}L_{p2}C_{ds}$ term becomes smaller and the magnitude of the denominator decreases, thereby increasing the magnitude of *Z*_21_ and improving the power gain. Even in this case, if L_p2_ becomes too large, the $1+s^{2}L_{p2}C_{ds}$ term increases in the negative direction and the power gain decreases. Therefore, it is important to choose an appropriate L_p2_ value between the two cases. 

The above equations can be used as a basis for the initial design. Approximate initial values of feedback inductances (L_p1_ and L_P2_) can be obtained by substituting the known model parameters of the transistor into the equations. However, the actual optimal value can be obtained through the iterative CAD simulation that more accurately reflects the design environment.

Figure 10a,b show the simulated CG according to the values of L_p1_ and L_p2_ at high frequency. As expected from Equations (2) and (7), CG changes according to the values of L_p1_ and L_p2_, and the optimal CG is shown when L_p1_ = 120 pH and L_p2_ = 30 pH. Although we did not calculate the equation when both inductors are simultaneously included because the formula becomes very complicated, some tuning is required. This can be readjusted from the initial value through simulation. However, it did not change much.

### 2.4. Final Distributed Mixer Design 

Figure 11 shows the circuit schematic of a distributed cascode mixer. It is designed using a commercial 130 nm CMOS process and Keysight’s advanced design system (ADS) software program. In this work, a single-ended design is adopted for a distributed mixer. A differential type is easy to obtain a ground at a high frequency by creating a virtual ground and is advantageous for common mode noise and spurious signal removal. In particular, when other RF systems are configured as a differential structure, it is a structure that is good for integration by interworking. On the other hand, when used independently or when other RF systems are single-ended, a balun that branches an input signal to a differential one and combines the differential signal into one output signal is required. In the differential type, additional loss occurs due to the balun, and power consumption increases and the chip size increases due to an increase in the number of transistors used. When implementing a wideband mixer, there should be a wideband balun. On the other hand, the single-ended type does not require a balun, so the structure is simple, the high frequency loss is small, and the number of transistors used is small, so power consumption is small compared with the differential type. Therefore, it is suitable for application to a distributed structure to obtain broadband characteristics. However, since many RF front-end systems are currently composed of a differential type, we will try to design the distributed mixer proposed in this study as a differential type in the next study.

The distributed mixer consists of three stages, and both M1 and M2 of the cascode cell use a 2 (number of gate fingers) × 15 μm (unit gate finger width) n-channel enhancement-type metal-oxide-semiconductor (NMOS). The RF and LO signals are distributed to each cascode mixer cell through the gate lines of M1 and M2, respectively. Then, the RF signal is applied to the gate of M1, the LO signal is applied to the gate of M2, and the IF signal is output to the drain of M2. The IF signal is summed and output through the drain lines of each cascode mixer cell. Both the gate line and drain line are designed as a coplanar waveguide (CPW)-type transmission line with low high-frequency loss and are easy to implement in a monolithic microwave integrated circuit (MMIC), and the line length is adjusted according to the inductance calculated in the previous section [10]. To implement inductive feedback, narrow inductive lines are added to each node of the cascode cell as shown in the colored part in Figure 11. Actually, as shown in Figure 10b, since the optimum value of L_p2_ is as small as 30 pH, it is implemented as a thin and short line connecting the drain of M1 and the source of M2. C_RF_ and C_LO_ are used as DC block capacitors, and C_IF_ is connected to the shunt to filter the RF or LO signal and pass the IF signal. By using R_RF_ and R_LO_ resistors, the gate biases of V_GG1_ and V_GG2_ of each transistor are supplied through the gate lines. RF gate lines and LO gate lines are terminated respectively with R_gt1_, C_gt1_ and R_gt2_, C_gt2_ for wideband matching. The V_DD_ is designed to be supplied by connecting an external bias tee to the IF port.

Figure 12 shows the distributed cascode mixer layout according to the schematic in Figure 11. In the center of the layout, three cascode mixer cells are included. For a compact layout, the transistors of the cascode mixer are placed facing each other as shown on the right side of Figure 12, and the drain and source directions are reversed so that the drain line can be exited through the center. At this time, metal vias are used for the line-to-line connection and transistor-to-transistor connection, and parasitic inductance and parasitic capacitance due to this can affect the high-frequency performance. It is necessary to reflect this effect in the simulation and examine the sensitivity according to the actual layout. The sensitivity variables are the transistor model variation, parasitic inductance by via interconnection near transistors, and the presence or absence of a tie-down diode added to satisfy the antenna rule. We simulated the conversion gain in each case and compared the variation of CG to see if there are any critical factors in performance. As a result, the CG decreased by up to 5 dB at frequencies above 60 GHz depending on the combination. 

Figure 13a,b show the simulated CG and return loss according to frequency of the distributed cascode mixer, respectively. The simulation conditions are −30 dB for RF power, 4 dBm for LO power, 1.5 V for V_DD_, and 0.7 V for V_GG1_ and V_GG2_. The IF is set near 1 GHz, and the IF matching is implemented by connecting an external 30 nH inductor in series so that the chip size does not increase. As shown in Figure 13a, the CG is approximately −3 to 3 dB from 4 to 65 GHz. A peak value of 2.8 dB is obtained at 11 GHz, and the CG is the lowest at –3.29 dB at 65 GHz. As shown in Figure 13b, the RF return loss is more than 10 dB from approximately 12 to 90 GHz, and the LO return loss is more than 10 dB from approximately 30 to 90 GHz. The IF signal shows the return loss of 18.5 dB at 1 GHz. 

The third input intercept point (IIP3) is simulated to examine the linearity of the designed mixer. IIP3 can be calculated through the third-order intermodulation distortion (IMD3) output by applying a two-tone RF signal with an interval of 10 MHz. Figure 14a shows the fundamental signal (fund), IMD3 signal (third), and the difference (imd) between the two signals obtained from the power sweep of the two-tone RF signal at 20 GHz. As shown in Figure 14b, IIP3 can be calculated from the simulated imd. The calculated IIP3 is about −2 to 12 dB from 4 to 66 GHz. Unfortunately, IIP3 was not measured because the measurement is not equipped.

## 3. Fabrication and Measurement

The proposed distributed cascode mixer was fabricated using a commercial 130 nm CMOS process with 1 poly and 8 metal layers. The cut-off frequency (f_T_) and maximum oscillation frequency (f_MAX_) of NMOS transistors provided by this process technology are about 90 and 100 GHz, respectively. Figure 15 shows the chip photograph of the fabricated mixer. The fabricated chip size is 0.8 × 0.7 mm^2^ and the active area without pads is 0.45 × 0.55 mm^2^.

Figure 16a,b show the on-wafer probing measurement setups of the fabricated mixer. The CG measurement setup is configured as shown in Figure 16a. RF and LO signals are generated and applied to the DUT chip by RF probes using the signal generator and Gunn source. An attenuator is used to sweep the LO power and a coupler monitors the actual LO power. DC bias is supplied to the mixer using a DC probe and bias tee, and the output IF signal is connected to a printed circuit board (PCB) with a 30 nH chip inductor for IF matching through the RF probe and measured with a spectrum analyzer. 

The return loss and isolation measurement setup is configured as shown in Figure 16b. A vector network analyzer (VNA) generates two-port small signals for each frequency, applies them to the DUT chip through an RF probe with LO and RF pads, and measures the reflected S-parameter signals using the VNA. A DC probe and bias tee are used to supply DC bias to the mixer, and the IF port is terminated with 50 ohm through the RF probe.

Figure 17 shows the measured CG versus RF frequency of the fabricated mixer. The fabricated mixer is measured under two bias conditions. The bias condition 1 is V_DD_ = 1.5 V, V_GG1_ = 0.6 V, and V_GG2_ = 0.7 V and the bias condition 2 is V_DD_ = 1.5 V, V_GG1_ = 0.5 V, and V_GG2_ = 0.5 V. The supplied RF power is −30 dBm, the LO power is 8–10 dBm, and the IF is 650 MHz. To obtain an optimal CG, the bias is slightly tuned, and the measured IF is slightly down-shifted by changing the matching element value. Moreover, the LO power is increased compared with the simulation result, which is presumably because the gate resistance of the transistor model used was underestimated in the simulation. In general, the larger the gate resistance, the more LO power is required [32,33]. Under bias condition 1, the measured CG is −2.9–3.1 dB at 4–30 GHz as shown in Figure 17, and at 54–66 GHz, it is −1.9 to 0.4 dB. The total DC current is 6–8 mA. Measurements are omitted in some bands because our measurement setup cannot generate enough LO power from 30 to 54 GHz. Compared with the simulation result (see the dotted line in Figure 17), similar results are generally shown within 4 dB. Around 30 GHz, the CG decreases more than the simulation. The reason is thought to be that the effect of inductive feedback was overestimated during the simulation. Conversely, the CG increases more than that of the simulation at 54–66 GHz, which is considered to result from the overestimation of the loss due to the high-frequency via model.

This time, the CG is measured in the low-power mode with a lower gate bias. As shown in Figure 17, when V_GG1_ and V_GG2_ are equally lowered to 0.5 V under bias condition 2, the current consumption is 3 mA and the measured CG is −7.97 to −1.47 dB at 4–30 GHz, lower than before, and −3.0 to 2.9 dB at 54–66 GHz, which is an increase from the previous one. At 65 GHz, the peak value is 2.9 dB, and at 30 GHz, it is the lowest at −7.97 dB. At low frequencies below 30 GHz, the decrease in g_m_ due to a low gate bias seems to have a greater effect on the CG.

Figure 18a,b represent the measured RF and LO return loss under bias conditions of V_DD_ = 1.5 V, V_GG1_ = 0.6 V, V_GG2_ = 0.7 V and V_DD_ = 1.5 V, V_GG1_ = 0.5 V, V_GG2_ = 0.5 V, respectively. The return loss does not change significantly depending on the bias condition. The RF return loss shows a wideband return loss of 10 dB or more from 10 to 67 GHz, but the LO return loss is more than 10 dB from 40 to 60 GHz only. Compared with the simulation result in Figure 13b, the RF return loss is similar to the simulation result, but the LO return loss is presumed to be worse because the pole is not formed at 80 GHz owing to the actual process error and model inaccuracy. 

Figure 19a,b represent the measured LO-RF isolation under bias conditions of V_DD_ = 1.5 V, V_GG1_ = 0.6 V, V_GG2_ = 0.7 V and V_DD_ = 1.5 V, V_GG1_ = 0.5 V, V_GG2_ = 0.5 V, respectively. As shown in Figure 19a,b, the LO-RF isolation is 15 to 70 dB at 0.2–67 GHz, which is the full band of the measurement frequency. In particular, when V_GG1_ and V_GG2_ are lowered to 0.5 V, the mixer obtains the excellent LO-RF isolation characteristics of 20 dB or more in all bands. 

Depending on the application of the wideband mixer, a filter is required. For example, when the proposed wideband mixer is used in the 28 GHz band for 5G communication, a band pass filter can be used to prevent interference with other communication frequencies, such as LTE and WiGig, and the out-of-band rejection of 40 dB or more is required by referring to the 3rd generation partnership project—radio access network working group 4 (3GPP RAN4) standard. In addition, a low-loss broadband balun is required to integrate into a system with a differential structure.

Table 2 summarizes the performance comparison of distributed mixers reported recently. Ⅲ–Ⅴ based distributed mixers such as indium phosphide (InP) and metamorphic high-electron-mobility transistors (mHEMT) show the most excellent bandwidth performance due to superior f_T_ and f_MAX_ of device technology. However, even though a commercial CMOS process was used in this work, the proposed mixer can compete with the other distributed mixers in terms of performance such as CG, power consumption, and LO-RF isolation. The proposed mixer also shows excellent overall characteristics among silicon-based distributed mixers. Compared with the mixer described in [12], which is most similar to it in terms of process and topology, the proposed mixer shows better CG and LO-RF isolation by using the cascode structure with inductive feedback. Although the CG is lower than that of the mixer described in [20], it shows lower power consumption and better LO-RF isolation. The mixer described in [20] has process limitations in that both HBT and FET must be supported for topology implementation, but this work has the advantage that it can be implemented with a generalized CMOS process.

## 4. Conclusions

A wideband and low-power distributed cascode mixer is demonstrated using a commercial 130 nm CMOS process. The proposed distributed mixer improves CG while minimizing power consumption by applying a cascode structure to the drain pumped mixer with low-power operation. In addition, inductive feedback is added to the cascode structure to improve the CG at high frequencies. Thus, with a power consumption of about 10 mW, an average CG of 0 dB or more was obtained covering UWB, the 28 GHz 5G communication band, and up to the 60 GHz WiGig band. Moreover, LO-RF isolation over a minimum of 15 dB was obtained in a wide band without using a complex balanced structure. The proposed mixer can be a good solution for future communication transceiver systems that require wideband and low-power operation.

## Figures and Tables

**Figure 1 sensors-22-09022-f001:**
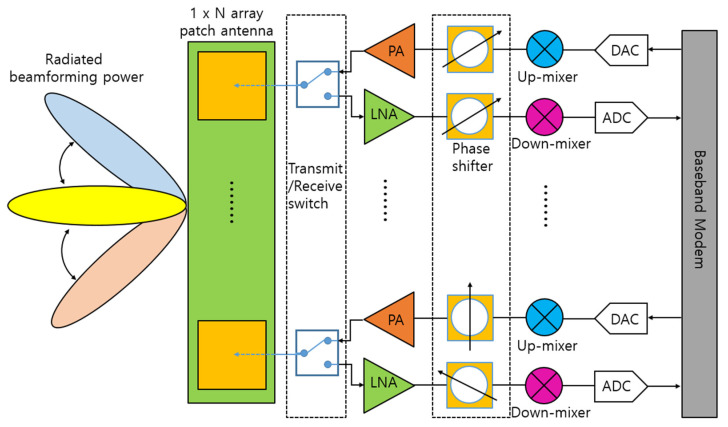
Block diagram of a 5G antenna module.

**Figure 2 sensors-22-09022-f002:**
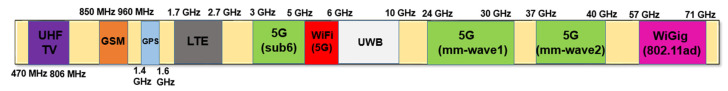
Frequency spectrum distribution in commercial use today.

**Figure 3 sensors-22-09022-f003:**
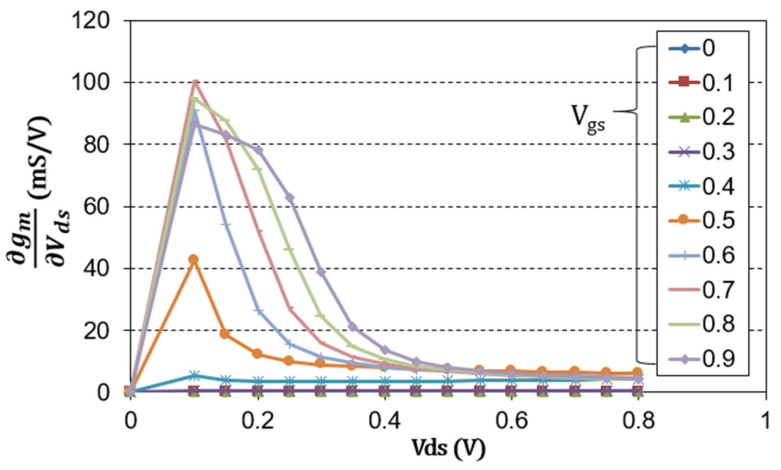
Simulated derivative of g_m_ with respect to V_ds_ according to V_gs_ and V_ds_.

**Figure 4 sensors-22-09022-f004:**
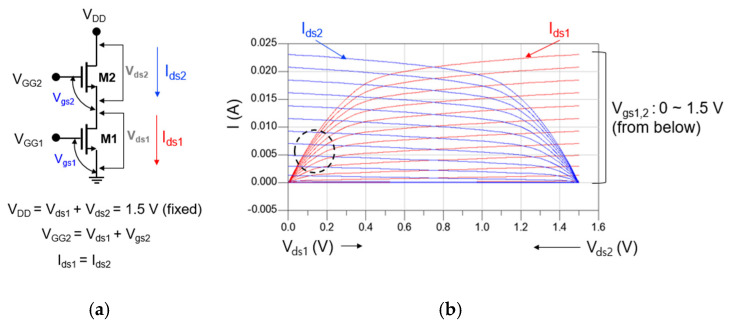
(**a**) Circuit schematic of the cascode mixer and (**b**) DC-IV of each transistor constituting the cascode structure.

**Figure 5 sensors-22-09022-f005:**
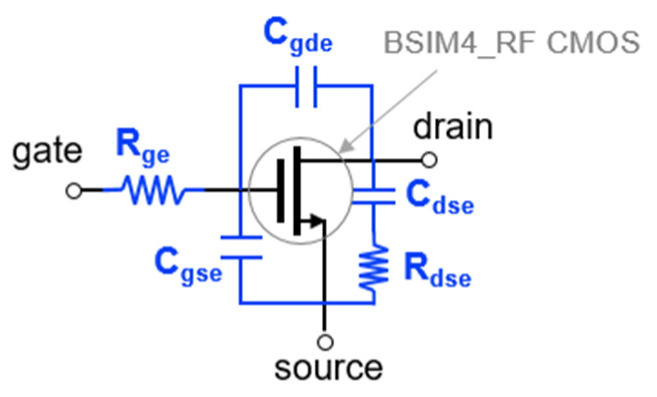
Macro model equivalent circuit of RF CMOS used in this work.

**Figure 6 sensors-22-09022-f006:**
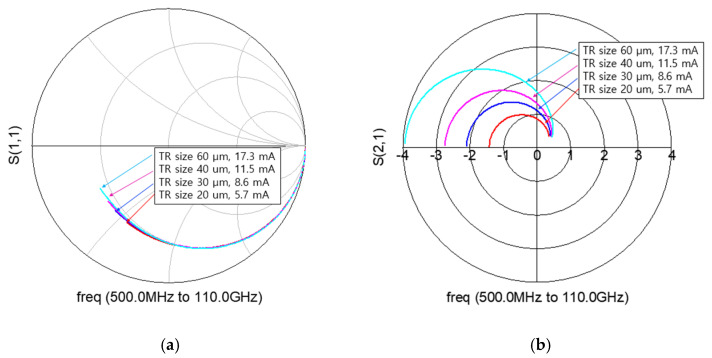
Comparison of (**a**) S11 and (**b**) S21 according to transistor (TR) size with current consumption.

**Figure 7 sensors-22-09022-f007:**
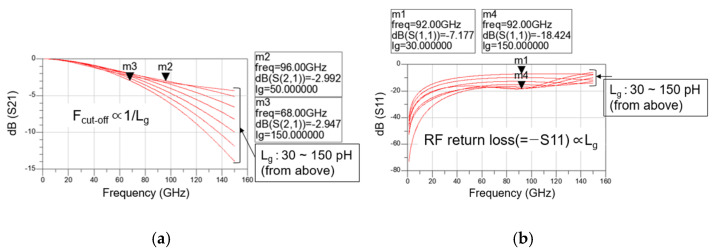
Simulated magnitudes of (**a**) S21 and (**b**) S11 in the total gate lines of a distributed cascode mixer versus RF according to gate line inductance.

**Figure 8 sensors-22-09022-f008:**
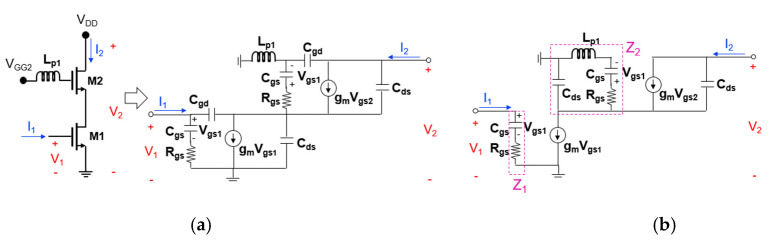
Transistor equivalent circuits of the cascode mixer when the first positive feedback inductor (L_p1_) is added to the gate node of the upper transistor ((**a**) two-port representation and (**b**) simplified one).

**Figure 9 sensors-22-09022-f009:**
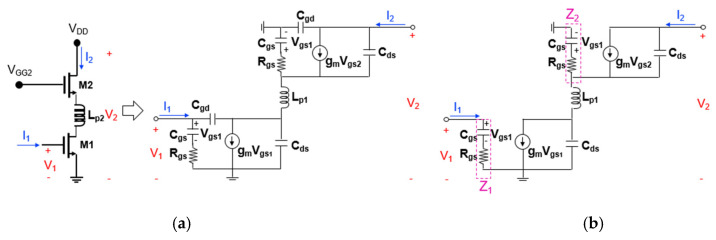
Transistor equivalent circuit of the cascode mixer when the second positive feedback inductor (L_p2_) is added to the source-drain connection between M1 and M2 ((**a**) two-port representation and (**b**) simplified one).

**Figure 10 sensors-22-09022-f010:**
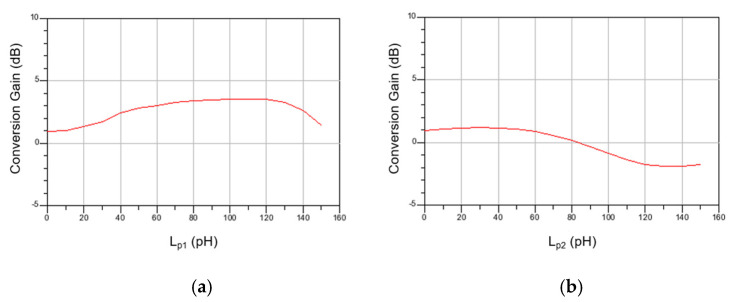
Simulated CG at high frequency according to (**a**) L_p1_ and (**b**) L_p2_.

**Figure 11 sensors-22-09022-f011:**
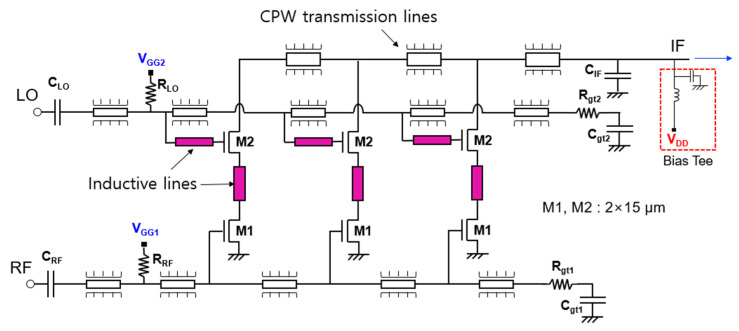
Circuit schematic of distributed cascode mixer.

**Figure 12 sensors-22-09022-f012:**
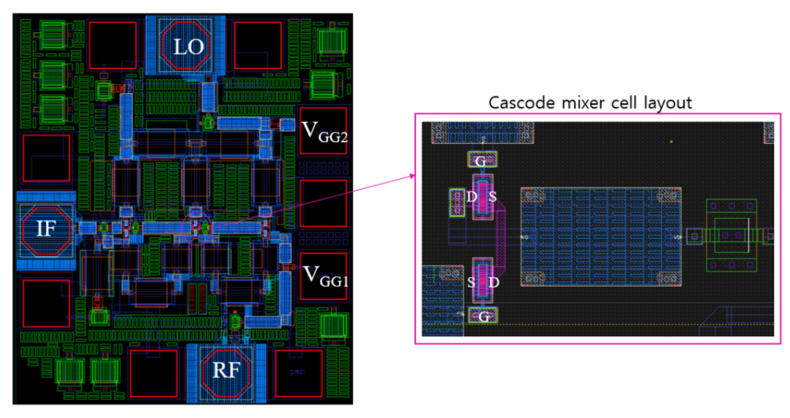
Layout of distributed cascode mixer.

**Figure 13 sensors-22-09022-f013:**
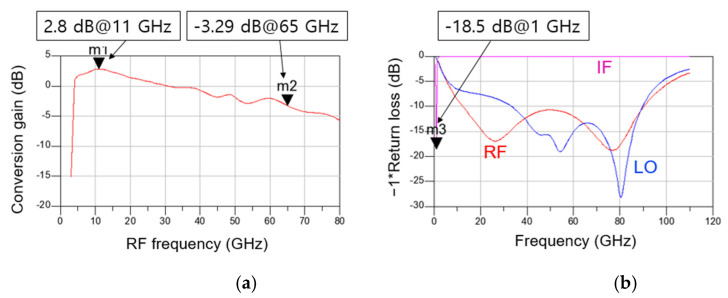
Simulated (**a**) CG versus RF frequency and (**b**) return loss versus frequency of the distributed cascode mixer.

**Figure 14 sensors-22-09022-f014:**
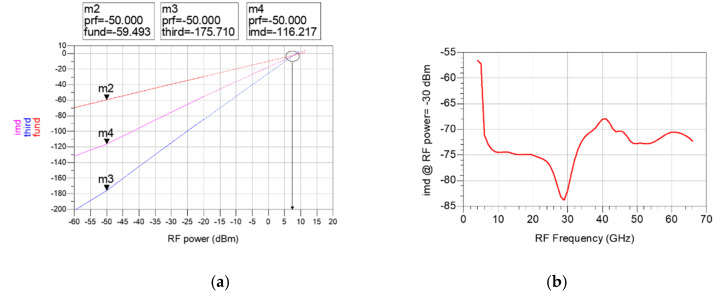
(**a**) Fundamental signal (fund), IMD3 signal (third) and the difference between the two signals (imd) according to the power sweep of the two-tone RF signal at 20 GHz (**b**) simulated imd according to RF Frequency (RF power = −30 dBm).

**Figure 15 sensors-22-09022-f015:**
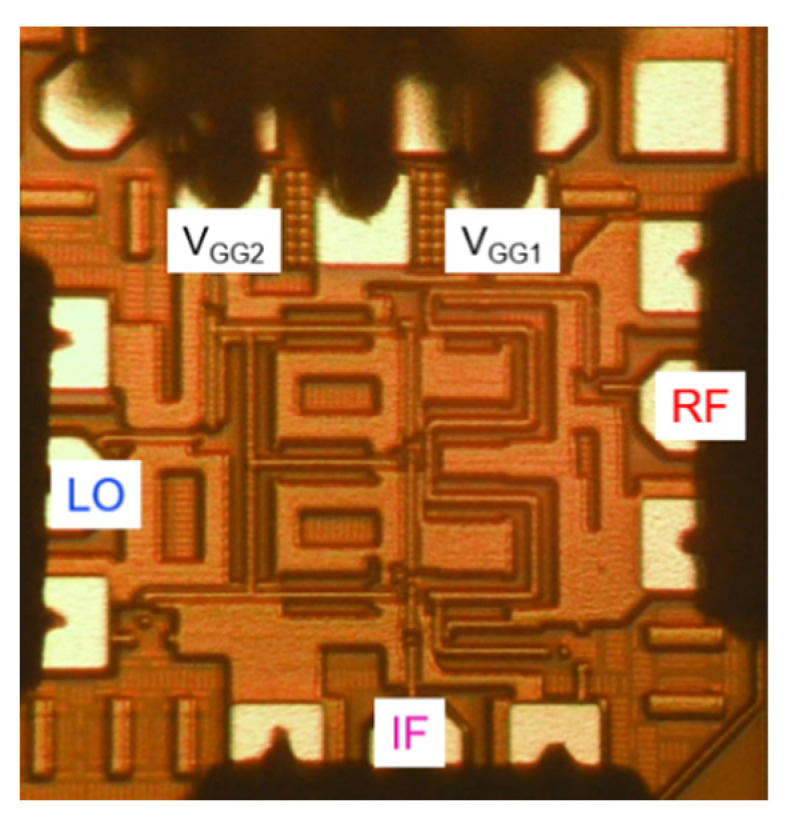
Chip photograph of the fabricated mixer (size: 0.8 × 0.7 mm^2^).

**Figure 16 sensors-22-09022-f016:**
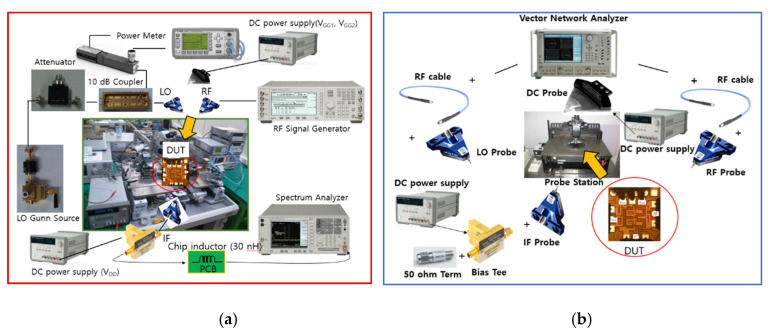
Measurement setup of the fabricated mixer ((**a**) CG, (**b**) return loss and isolation).

**Figure 17 sensors-22-09022-f017:**
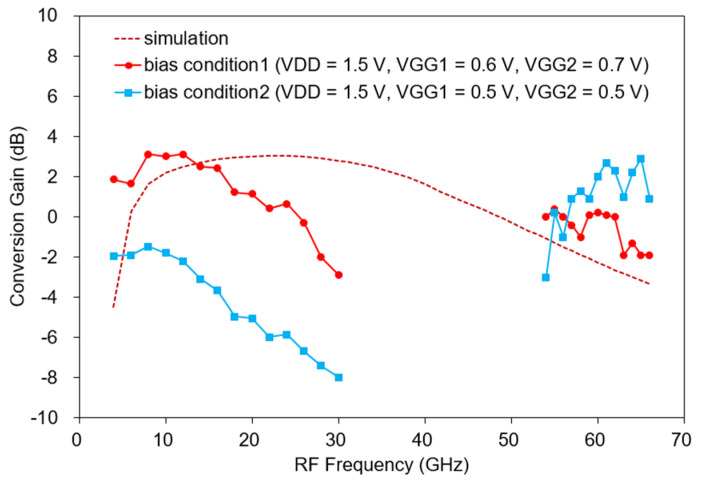
Measured CG versus RF (bias condition1: V_DD_ = 1.5 V, V_GG1_ = 0.6 V, V_GG2_ = 0.7 V, bias condition2: V_DD_ = 1.5 V, V_GG1_ = 0.5 V, V_GG2_ = 0.5 V, LO power = 8–10 dBm, RF power = −30 dBm, IF = 650 MHz).

**Figure 18 sensors-22-09022-f018:**
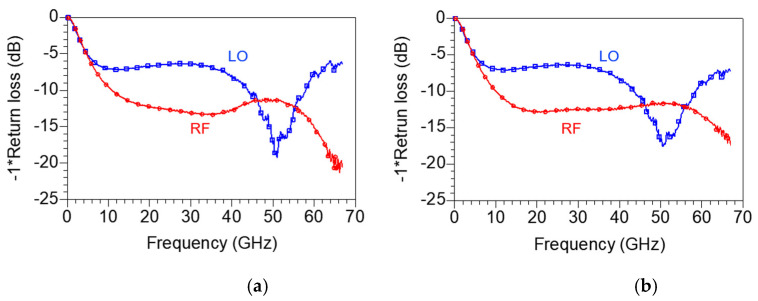
Measured RF and LO return loss under bias conditions of (**a**) V_DD_ = 1.5 V, V_GG1_ = 0.6 V, V_GG2_ = 0.7 V and (**b**) V_DD_ = 1.5 V, V_GG1_ = 0.5 V, V_GG2_ = 0.5 V.

**Figure 19 sensors-22-09022-f019:**
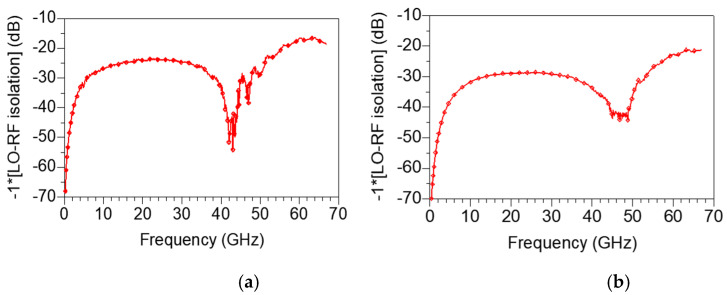
Measured LO-RF isolation under bias conditions of (**a**) V_DD_ = 1.5 V, V_GG1_ = 0.6 V, V_GG2_ = 0.7 V and (**b**) V_DD_ = 1.5 V, V_GG1_ = 0.5 V, V_GG2_ = 0.5 V.

**Table 1 sensors-22-09022-t001:** Simulated 3-dB cutoff frequency (F_cut-off_) according to the stage number when the transistor size is fixed to 30 μm.

Stage Number	1	2	3	4
F_cut-off_ (GHz)	125	81	63	54

**Table 2 sensors-22-09022-t002:** Performance comparison of distributed mixers.

Reference	Frequency (GHz)	Technology	CG (dB)	LO Power (dBm)	LO-RF Isolation (dB)	DC Power Consumption (mW)	Chip Size (mm^2^)	Topology
[12]	0.8–77.5	130 nm CMOS	−5.5 ± 1.0	10	>13	0	0.388	Drain mixer
[13]	1–170	35 nm mHEMT	−4.2–1.0	−1	>20	180	1.5	Source feedback mixer
[19]	0–194	250 nm InP HBT	−4.4–5.7	2	-	125	0.8	Single balanced mixer
[20]	2–67	180 nm SiGe BiCMOS	1.7–4.8	0	>10	17.5	0.42	Darlington cell
[21]	5–45	180 nm CMOS	−12.2 ± 1.0	8	33–47	1.4	0.66	Cascoded switching pairs
[22]	0.1–40	150 nm GaAs pHEMT	−6.0 ± 1.0	10	>15	0	0.9	Symmetric drain mixer
[23]	5.4–21.8	250 nm GaAs pHEMT	−4.0–−7.4	2	>23.5	0	6.4	Balanced drain mixer
This work	4–30, 54–66	130 nm CMOS	−2.8–3.3	8–10	>15	10–12	0.56	Cascode mixer with inductive feedback
			−8.0–2.9	8–10	>20	4.5		

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
