# Peer review of "A Wideband and Low-Power Distributed Cascode Mixer Using Inductive Feedback"

_sensors, 2022, doi:10.3390/s22229022_

Round 1
Reviewer 1 Report
Distibuted wideband mixer was proposed in this paper.
Although silimar techniques were used in several stacked PA papers, this is the first demonstration applied to active mixer paper.
The analysis and measurement were both well presented. Howeverm authors should mention that why the wideband mixer is need. wideband mixer can accept many out-of-band interferers without filtering. Also, mostly resevers adopt differential structure in rx chain, unlike the proposed single-ended mixer.
Font size and editing should be revised overally.
Author Response
Dear associate editor and reviewers,
I would like to thank the reviewers for their valuable comments. Here are my answers to the reviewers' comments.
Although the revision period wasn't long, I did my best to incorporate all the comments into the revision as best I could. Please, take good care of me. Although the study is still lacking, I look forward to publishing it in your journal.
Thank you.

Reviewer 2 Report
1. The manuscript comments on wireless communication systems' importance and the presence of a mixer in these systems. Please, comment on the most popular mixer circuit topologies and compare them with benefits and drawbacks to highlight the proposed circuit.
2. The paper shows a single-ended topology. However, the relevant RF mixers upconvert /downconvert a fully-differential signal into a single-ended one. Comment on the benefits or disadvantages of converting to a single-ended version.
3. This work presents the mixer design from the circuit design point of view. However, the natural approximation is the Radio Frequency context. Therefore the cut-off frequency is important but the transistors' Smith graph is more relevant. Please include this information instead of table 1-2 and concentrate the information.
4. Lines 117-120 mention the transistors' impedance can be simplified. This information is not correct, the device's impedance is very important and should not simplify with concentrated models. The Smith card helps in understanding the important parameters for the mixer design.
5. Please, specify the RF model of the MOS devices. Section 2.3 mentions the gate line model is an inductance. It is difficult if this model is not supported with experiments. Usually the foundry provides the RF model. Please, make further comments.
6. The small signal representation in Fig. 7 is too simple for the RF analysis.
7. Equation (1) is not correct. Please, use the Two-Port representation in terms of voltages and currents.
8. The comments in lines 148-160 refer to a scattering parameter of 7dB, which should be -7dB.
9. The paragraph in Lines 235-242 expresses the design considerations for a reliable and robust design. The Section mentions sensitivity but there is no clear evidence of robustness with a Monte Carlo and or Process, Voltage, and Temperature changes.
10. The results Section needs an overall improvement. The necessary design and characterization parameters of an RF mixer are the conversion factor (dB) and IIP3 (dBm). Please include this in your characterization and make further comments.
Locate the Figures at the top page.
Author Response

(The authors gave the same response as above.)
